# Chromatic-Aberration-Corrected Hyperspectral Single-Pixel Imaging

Ying Liu [1], Zhao-Hua Yang [1,*], Yuan-Jin Yu [2,*], Ling-An Wu [3], Ming-Yue Song [1] and Zhi-Hao Zhao [1]

1   School of Instrumentation and Optoelectronics Engineering, Beihang University, Beijing 100191, China
2   School of Automation, Beijing Institute of Technology, Beijing 100081, China
3   Institute of Physics, Chinese Academy of Sciences, Beijing 100190, China
*   Correspondence: yangzh@buaa.edu.cn (Z.-H.Y.); yuanjin.yu@bit.edu.cn (Y.-J.Y.)

**Abstract:** With the emerging development of hyperspectral single-pixel imaging (SPI) systems, the trade-off between the simplicity of optical structure and the correction of chromatic aberration is now an essential factor to be considered. To address both issues simultaneously, we propose a chromatic-aberration-corrected hyperspectral single-pixel imaging scheme, which is based on spectral measurement and dispersion correction. Its achromatism feature is evaluated by optical simulations and proof-of-concept experiments. Moreover, to overcome the shortcomings of traditional algorithms, a new adaptive iterative algorithm is employed, which can further optimize image quality. The results demonstrate that both dispersion and noise in our system are significantly reduced. Taking the position coordinate variance as a figure of merit, we have realized an order of magnitude improvement in the lateral chromatic aberration over the spectral range of 400–780 nm compared to that in conventional hyperspectral SPI. Meanwhile, the contrast-to-noise ratio in our system is enhanced on average by 3 dB. To the best of our knowledge, this is the first such demonstration, and the technique presents possibilities for future integrated applications of high spatial/spectral resolution over the entire visible range, and the system has the potential to be scaled down for future integrated applications.

**Keywords:** single-pixel imaging; hyperspectral; chromatic aberration correction; optical system

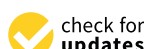



## 1. Introduction

The first recorded attempt at single-pixel imaging may be traced back to 1884, when Paul Nipkow patented the flying-spot camera, using a Nipkow disk to modulate the light source [1]. After years of development, contemporary single-pixel imaging (SPI) has advanced in parallel with computational ghost imaging (GI) [2,3]. The basic idea is to modulate the reflected/transmitted beam illuminating an object with a sequence of patterns and record the corresponding total intensity values with a single-pixel (bucket) detector, then recover the image through convolution. The patterns are generally pre-ordered and generated by a programmable spatial light modulator (SLM). The measured data, combined with the prior information of the illumination speckles, constitute the essential elements of a given computational reconstruction algorithm.

Currently, there is a variety of reconstruction algorithms. Ferri et al. first proposed differential GI in 2010 [4], which has been widely adopted for its intrinsic advantage of anti-noise interference. Later, many other algorithms were proposed, for instance, normalized GI [5], high-order GI [6], correspondence GI [7] and self-evolving GI [8].

In single-pixel imaging, only the total intensity values need to be measured, which makes it particularly suitable for applications in low visibility conditions, such as in remote sensing [9,10] and underwater imaging [11]. Formerly, single-pixel sensing operated on a single wavelength. When the process is extended to a wide spectral range, chromatic aberration occurs since the same material exhibits different refractive indices at different

wavelengths. This type of deviation would restrict the further enhancement of the spatial resolution of a broadband single-pixel imager. More severely, the larger the spectral range, the larger the chromatic aberration would be. Currently, known solutions, including temporal multiplexing [12] and frequency-division multiplexing [13], which are akin to introducing optical bandpass filters, predominantly suffer from the redundancy of optical structure and restricted spectral resolution. Schemes of coded aperture snapshot spectral imagers [14–16] could solve the problem, but with a significantly increased complexity of the algorithms.

Hyperspectral imaging has been implemented based on SPI with a flat-field grating [17–19]. However, these schemes paid little attention to chromatic aberration since they were basically proximity range detection. To minimize the impact of chromatic aberration and to reduce structural vulnerability, spectral anti-aliasing with as simple as possible optical arrangements is required. In traditional spectral systems, this is achieved by dispersing the incident polychromatic light before the imaging section with an independent set of devices for each waveband, but these systems are always heavy and cumbersome.

In this article, we propose a solution based on spectral measurement and dispersion correction, which we call chromatic-aberration-corrected hyperspectral single-pixel imaging (CHSPI). A spectrometer is used as the single-pixel sensor, so as to measure the spectrum of the reflected light. A closed-circuit television camera lens (later referred to as the TV lens) with a double Gaussian structure is adopted instead of a single thin lens. To eliminate the ripple noise when the object is imaged in a real light field and reconstructed by a traditional SPI algorithm, an adaptive and iterative correction algorithm is formulated. Simulations and a series of proof-of-concept experiments are conducted to test the achromatic capability of our scheme. As far as we are aware, this is the first demonstration of chromatic aberration correction in broadband hyperspectral SPI. The lateral chromatic aberration is suppressed by an order of magnitude over 400–780 nm in terms of pixel coordinate variance. Compared with previous schemes, our system can achieve high-contrast achromatic hyperspectral imaging over a broad range while the structure is lightweight and stable—characteristics that are particularly desirable in remote sensing applications.

## 2. System Model and Theoretical Derivation

### 2.1. Principle of Hyperspectral Differential Ghost Imaging

In our scheme, a differential GI-based algorithm was selected to reconstruct the image while suppressing the noise [4]. In this linear non-iterative method, a variable approximate to the average intensity of the light field is subtracted from each pixel. In this way, the effective signal of the bucket detector just reflects the fluctuations of the light so that the noise is significantly suppressed and the signal-to-noise ratio (SNR) of the reconstructed image is improved.

The basic principle of SPI is shown in Figure 1, where SLM is a spatial light modulator.

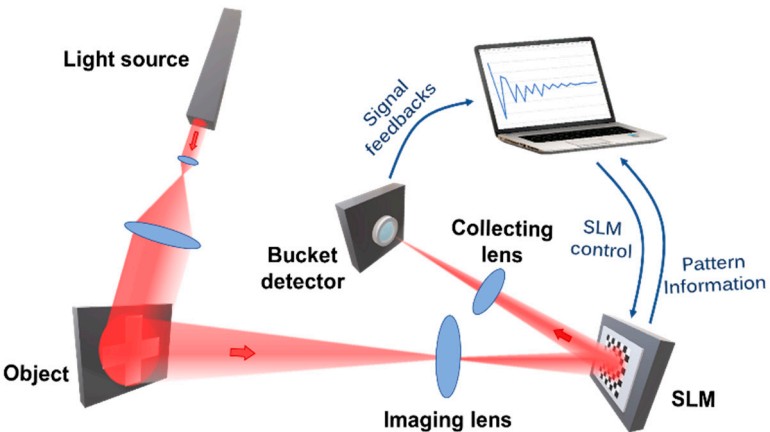

**Figure 1.** Principle of single-pixel imaging.

In traditional SPI, the image is restored by correlating the light field modulation function $\mathbf{I}(x)$ with the bucket signal $\mathbf{S}$:

$$\mathbf{S} = \int_{A_S} \mathbf{I}(x)\mathbf{R}(x)\mathrm{d}x \tag{1}$$

where $\mathbf{R}(x)$ is the (intensity) reflection function of the object, and $A_S$ is the plane on which the image is modulated. The retrieved image is thus:

$$\mathbf{G}(x) = \langle \delta\mathbf{S} \cdot \delta\mathbf{I}(x) \rangle \tag{2}$$

where $\langle \ldots \rangle = \frac{1}{N}\sum_{i=1}^{N} (\ldots)$ denotes the ensemble average and $\delta\mathbf{I}(x) = \mathbf{I}(x) - \langle\mathbf{I}(x)\rangle$. In differential GI, the fluctuating part of $\mathbf{R}(x)$ is defined as $\delta\mathbf{R}(x) = \mathbf{R}(x) - \overline{\mathbf{R}}$, where $\overline{\mathbf{R}} = \int \langle\mathbf{I}(x)\rangle \mathbf{R}(x)\mathrm{d}^2x / \int \langle\mathbf{I}(x)\rangle\mathrm{d}^2x$ is the average transmission function of the object. We define a new differential bucket signal $\mathbf{S}_1$ as:

$$\mathbf{S}_1 = \int_{A_S} \mathbf{I}(x)\delta\mathbf{R}(x)\mathrm{d}x \tag{3}$$

Suppose $N \times N$ is the number of pixels in each pattern and $K$ is the number of preset 2D spatial patterns ($K \leq N \times N$). Since the shot noise is negligible compared with the abundant detected photons, the known modulation matrix $\mathbf{M}(x) \in \mathbb{R}^{N \times N \times K}$ and the beam $\mathbf{I}(x) \in \mathbb{R}^{N \times N \times K}$, which is reflected by the SLM, are perfectly correlated variables; we can introduce a factor $\alpha$ to describe the correlation as $\mathbf{I}(x) = \alpha\mathbf{M}(x)$. At the same time, we can set a virtual bucket signal $\mathbf{S_R} = \int \mathbf{M}(x)\mathrm{d}^2x$ to simplify the expression. Notice that $\overline{\mathbf{R}}$ can be expressed in terms of $\langle\mathbf{S_R}\rangle$ as:

$$\overline{\mathbf{R}} = \frac{1}{\alpha} \frac{\langle\mathbf{S}\rangle}{\langle\mathbf{S_R}\rangle} \tag{4}$$

In this way, $\mathbf{S_1}$ can be measured from the following operative form

$$\mathbf{S}_1 = \mathbf{S} - \frac{\langle\mathbf{S}\rangle}{\langle\mathbf{S_R}\rangle}\mathbf{S_R} \tag{5}$$

The image is then recovered by substituting $\mathbf{S}$ with $\mathbf{S_1}$ in Equation (2), namely,

$$\begin{aligned} \mathbf{G}_1(x) &= \langle \delta\mathbf{S}_1 \cdot \delta\mathbf{I}(x) \rangle \\ &= \langle \mathbf{S} \cdot \mathbf{I}(x) \rangle - \frac{\langle\mathbf{S}\rangle}{\langle\mathbf{S_R}\rangle}\langle \mathbf{S} \cdot \mathbf{I}(x) \rangle \end{aligned} \tag{6}$$

In hyperspectral imaging, the above reconstruction of grayscale images corresponds to periodic sampling in the spectral dimension. More specifically, the detected intensity values over every 10 nm of the spectrum are summed as the "bucket detector value" $\mathbf{S}_i$ of the corresponding band I, from which the grayscale image $\mathbf{G}_{1i}(x)$ is reconstructed as in Equation (7) below:

$$\begin{aligned} \mathbf{G}_{1i}(x) &= \langle \delta\mathbf{S}_{1i} \cdot \delta\mathbf{I}(x) \rangle \\ &= \langle \mathbf{S}_i \cdot \mathbf{I}(x) \rangle - \frac{\langle\mathbf{S}_i\rangle}{\langle\mathbf{S}_{iR}\rangle}\langle \mathbf{S}_i \cdot \mathbf{I}(x) \rangle \end{aligned} \tag{7}$$

Based on the CIE 1931 Standard Colorimetric Observer, within the visible spectrum of 400–780 nm, the weighted sum of the images in each band is used to obtain the mixed color tristimulus image $\mathbf{X}(x)$, $\mathbf{Y}(x)$ and $\mathbf{Z}(x)$:

$$\mathbf{X}(x) = k\sum \mathbf{G}_{1i}(x) \cdot \overline{\mathbf{x}}(\lambda)\Delta\lambda \tag{8}$$

$$\mathbf{Y}(x) = k\sum \mathbf{G}_{1i}(x) \cdot \overline{\mathbf{y}}(\lambda)\Delta\lambda \tag{9}$$

$$\mathbf{Z}(x) = k\sum \mathbf{G}_{1i}(x) \cdot \overline{\mathbf{z}}(\lambda)\Delta\lambda \tag{10}$$

where $\overline{\mathbf{x}}(\lambda)$, $\overline{\mathbf{y}}(\lambda)$, and $\overline{\mathbf{z}}(\lambda)$ are the color matching functions, $k$ is the adjustment coefficient, and $\Delta\lambda$ is a fixed wavelength interval [20]. Then, by respectively plugging the three

matrices $\mathbf{X}(x)$, $\mathbf{Y}(x)$, and $\mathbf{Z}(x)$ into the three channels of the resultant RGB image, the entire reconstruction process can be completed.

### 2.2. Principle of Chromatic Aberration Correction

The refractive index of a given material varies with the wavelength of light. Therefore, in a typical single thin lens imaging system, the object is imaged at different locations when illuminated by polychromatic light; in other words, the image would be aliased into defocused speckles at any position [20], as shown in Figure 2. Thus, this chromatic aberration would decrease the accuracy of R(x) in Equation (1).

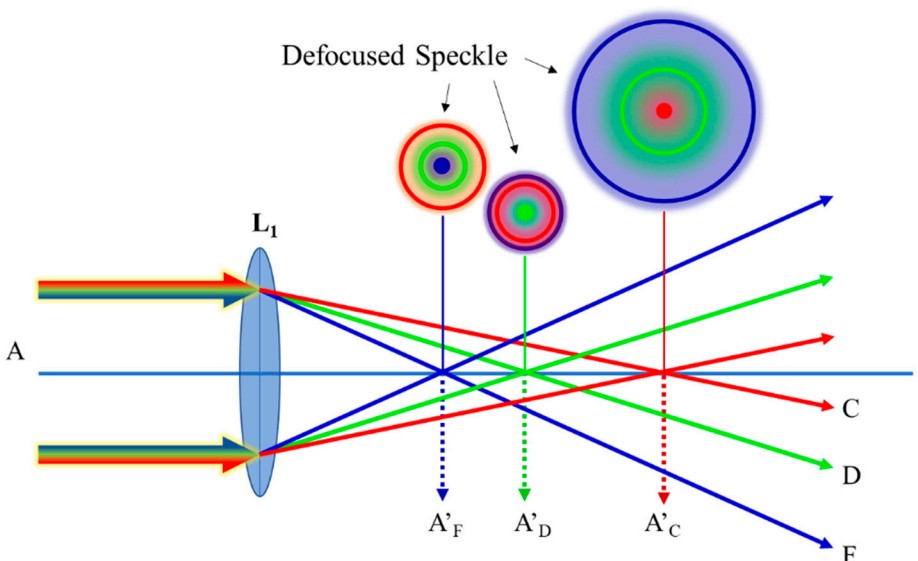

**Figure 2.** Schematic of chromatic aberration along the optical axis. A: object beam. Wavelengths of F, D, and C are 486 nm, 588 nm, and 656 nm, respectively.

To mitigate the problem, achromatic design is required. For a visual system, D light (588 nm) is generally chosen as the reference wavelength. The imaging positions of F light (486 nm) and C light (656 nm) are designed to coincide with the reference. The deviation between the imaging positions of two colors on the optical axis is called positional chromatic aberration, represented by

$$\Delta L'_{\text{FC}} = L'_{\text{F}} - L'_{\text{C}} = -\frac{\sum_i h^2 \frac{\varphi_i}{\nu_i}}{n'^2 u'^2} \tag{11}$$

where $h$ is the projection height of the object on each lens, $n'$ and $u'$ are the refractive index and aperture angle on the image side, and $\varphi_i$ and $\nu_i$ are the focal power and Abbe coefficient of each lens, respectively [20].

To eliminate the chromatic aberration, $\sum_i h^2 \frac{\varphi_i}{\nu_i}$ is expected to be zero. It can be seen

that a single lens cannot correct the chromatic dispersion because its focal power, $\varphi$, is always a non-zero constant but may be negative or positive depending on whether the lens is convex or concave, respectively. By selecting a suitable combination of lenses, $\sum_i h^2 \frac{\varphi_i}{\nu_i}$ can

be reduced to zero.

Our scheme uses broadband light to illuminate a linear array detector for detection, so eliminating chromatic aberration is a key issue. Meanwhile, considering the advantage of reducing vertical-axis aberration (such as distortion, etc.), the double Gaussian lens is used. The double Gaussian lens is a typical successful adoption of the above combination, as shown in Figure 3. Two meniscus lenses, one positive meniscus $L_1$ and another negative meniscus $L_2$, can reduce the spherical aberration and longitudinal chromatic aberration by

splitting one positive lens into two so that the curvature is spread over four surfaces instead of two, while the overall power of the lens is preserved. Subsequently, placing lenses $L_3$ and $L_4$ to face $L_1$ and $L_2$ symmetrically would notably diminish the vertical axis aberration, such as distortion. The vital step to correct the chromatic aberration is through replacing the negative meniscus lenses $L_2$ and $L_3$ with hyperchromatic lenses $L_{21}$, $L_{22}$ and $L_{31}$, $L_{32}$, respectively. A hyperchromatic lens is a cemented doublet lens with the same refractive index at the given wavelength but different dispersion, such that changing the radius of the curvature of the cemented surface does not alter the focal length of the primary wavelength but adjusts the chromatic aberration [21]. By arranging the above lenses in proper positions, the final double Gaussian lens can correct chromatic aberration.

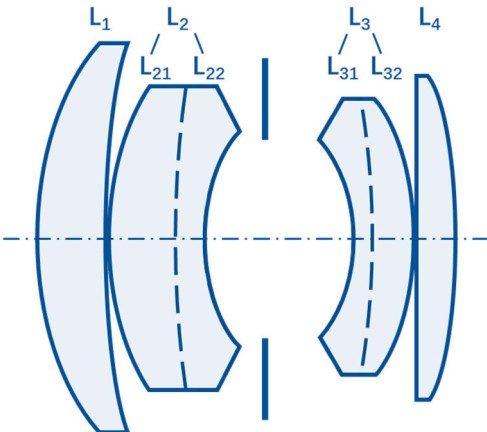

**Figure 3.** Double Gaussian structure.

## 3. Simulation

A series of simulations were performed to compare the capability to correct for chromatic aberration in conventional hyperspectral SPI (below referred to as HSPI) and our CHSPI systems. The focal length and aperture of the two groups were equally set to 50 mm and 25 mm, respectively. Figure 4 shows the general layouts, the lateral chromatic aberration curves, and the chromatic focal shift curves of the two systems. The lateral chromatic aberration is generally measured by the furthest deviation from the vertical axis in the field and the chromatic focal shift by the maximum difference between the focal positions in the given wavelength range. As the diagrams in Figure 4c,d illustrate, the maximal absolute value of lateral chromatic aberration in the CHSPI system is only 1/7 of that in the HSPI system over the wavelength range of 400–780 nm and the corresponding chromatic focal shift 1/15.

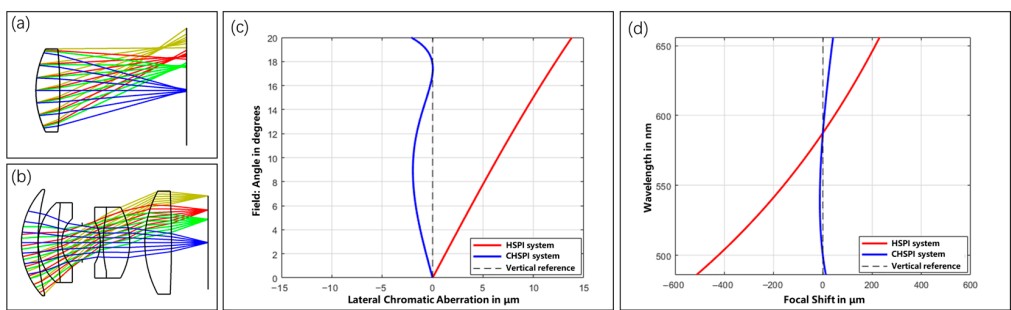

**Figure 4.** Simulation results of a single-lens system and TV lens system. Layout of imaging parts in (**a**) hyperspectral SPI (HSPI) and (**b**) chromatic-aberration-corrected hyperspectral single-pixel imaging (CHSPI). (**c**) Lateral chromatic aberration curve: field of view vs. lateral chromatic aberration. (**d**) The chromatic focal shift curve: wavelength vs. chromatic focal shift in the range from 486 nm of F-light to 656 nm of C-light.

## 4. Experimental Details

To verify the effectiveness of dispersion correction of our hyperspectral correlation imaging experimental system, we conducted an experiment consisting of a system based on the TV lens and a control group based on a single lens. The dispersion correction performance of the two systems was evaluated by comparing the relative deviations of the reconstructed images of different spectral bands at corresponding points in the recovered single-pixel images of the two groups under the same environmental conditions.

The schematic diagram and actual view of our experimental setup are illustrated in Figure 5. A broadband beam of light from a supercontinuum laser source (LEUKOS ROCK 400, 400–2400 nm) was expanded to illuminate the object, a $3 \times 3 \times 3$ Rubik's cube of size $54 \times 54 \times 54$ mm$^3$, which was then imaged by a TV lens onto a common type of SLM—a digital micromirror device (DMD) (ViALUX V-9501, $1920 \times 1080$) preprogrammed with a series of Hadamard matrices. After modulation, the beam was focused into a spectrometer (Ocean Optics FLAME-S-UV-VIS, 340–1100 nm). We employed 4096 speckle patterns generated under the cake-cutting basis Hadamard basis algorithm [22] for both the control and TV lens measurements.

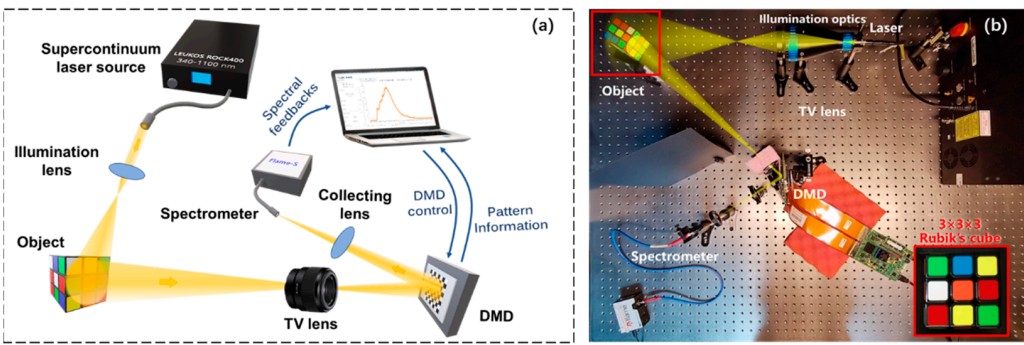

**Figure 5.** Schematic (**a**) and photograph (**b**) of the CHSPI setup.

## 5. Algorithm Optimization

In the process of reconstructing the grayscale image of each band, erroneous black points (zero points) are prone to appear at the upper-left edge, forming noise ripples, as shown in Figure 6a. For a picture with a homogeneous background and a contrasting object, these segments should have been part of and not darker than the background. As Equation (2) shows, the essence of GI computation is matrix multiplication. Under the full-sampling condition, the solution should be unique. Because the modulation matrix is a constant known parameter, only various unpredictable noises can explain this error. To minimize these errors while retaining the original information as much as possible, several methods are adopted. According to the ripple characteristics, the pixels adjacent to the error point are also prone to error, which excludes the traditional median or mean filtering methods of correction. Nor can frequency domain filtering be used since the ripples do not appear in all areas of the image. Therefore, on the basis of differential GI but to overcome its inadequacies, we designed a new adaptive iterative correction algorithm, as illustrated in Figure 6b.

In each iteration, the gray value of the error point is set as the global gray scale average to eliminate its impact on the image contrast before re-normalization. The error point always preferentially appears at the position of $(1, 2^i)$ or $(2^i, 1)(i = 1, 2, 3 \ldots \ldots)$. After each correction, its position moves from $(1, 2^i)$ to $(1, 2^{i-1} + 1)$ or from $(2^i, 1)$ to $(2^{i-1} + 1, 1)$ but never beyond the upper-left quarter of the image. We define the edge region as two line segments from $(1, 2)$ to $(1, N/2)$ and from $(2, 1)$ to $(N/2, 1)$ to simplify the computation. Each time, we search for the first zero point from left to right and from top to bottom. The whole process is finished when the first zero point no longer appears in the edge region. The SPI images of different spectral bands before and after correction are shown in Figure 6c. To quantitatively evaluate the image quality, we use the contrast-to-noise ratio

(CNR) instead of the conventional signal-to-noise ratio because we do not have an original object image for comparison. The CNR is defined as [23]:

$$\text{CNR} = 10 \log_{10}(S_o / \sigma_b) \tag{12}$$

where $S_o$ is the variance of the object area, and $\sigma_b$ is the variance of the background area. An average increase of more than 3 dB in Figure 6c clearly exemplifies the improvement in image quality by our algorithm.

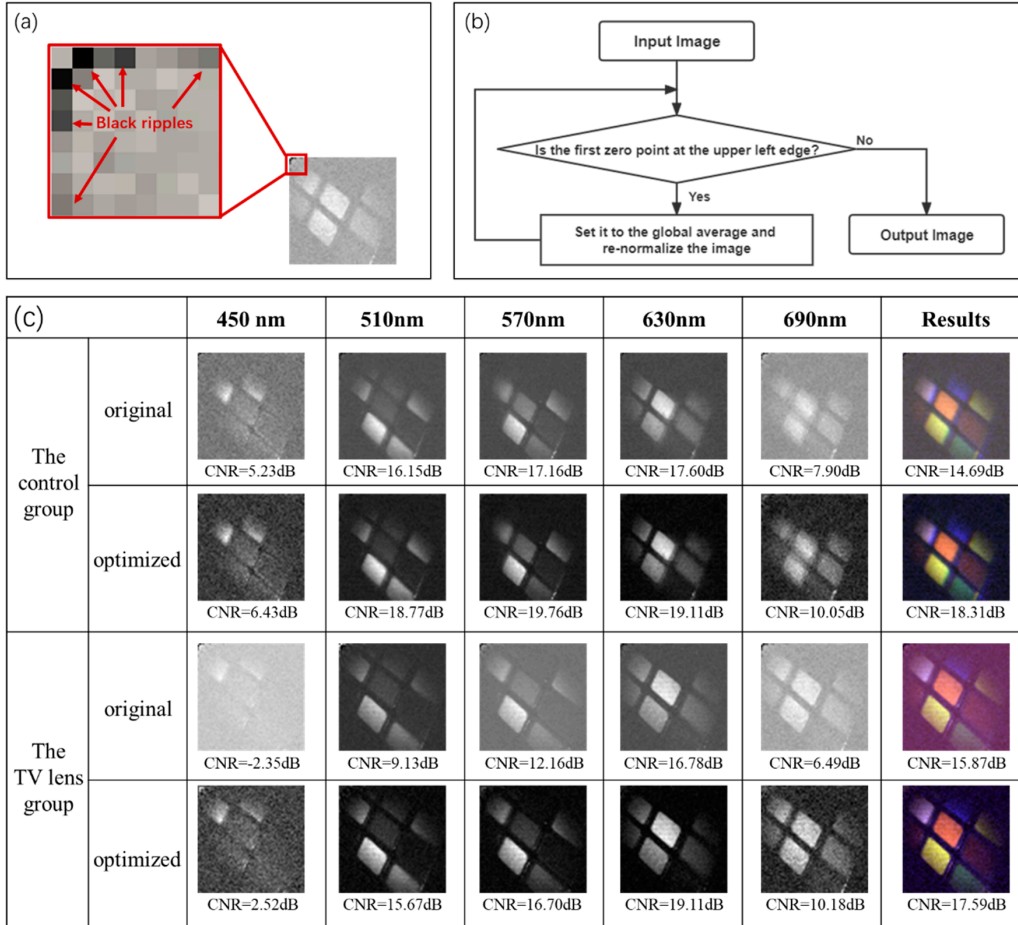

**Figure 6.** Proposed adaptive iterative correction algorithm. (**a**) The noise ripples (zero points) on the upper-left edge. (**b**) The adaptive iterative correction algorithm used to correct the erroneous black noise ripples. (**c**) Comparison of SPI images in different spectral bands before and after correction.

## 6. Results and Discussion

The optimized results of each wavelength band of hyperspectral SPI in the control and experimental groups are shown in Figure 7, where pseudo-coloring has been applied. In Figure 8, under the CIE1931 standard observer model, images of all spectral bands are fused to obtain the reconstructed color images. In the control group of Figure 8a, we can see the influence of lateral chromatic aberration, as images are only clear from 580–610 nm, elsewhere being blurred to varying degrees. By contrast, in the aberration-corrected images in Figure 8b, the block margins are sharply outlined and distinct from the background for all the bands; in addition, we can even discern the patchy reflections from the uneven surfaces of the colored squares, which are actually composed of glossy paper glued onto a black cube.

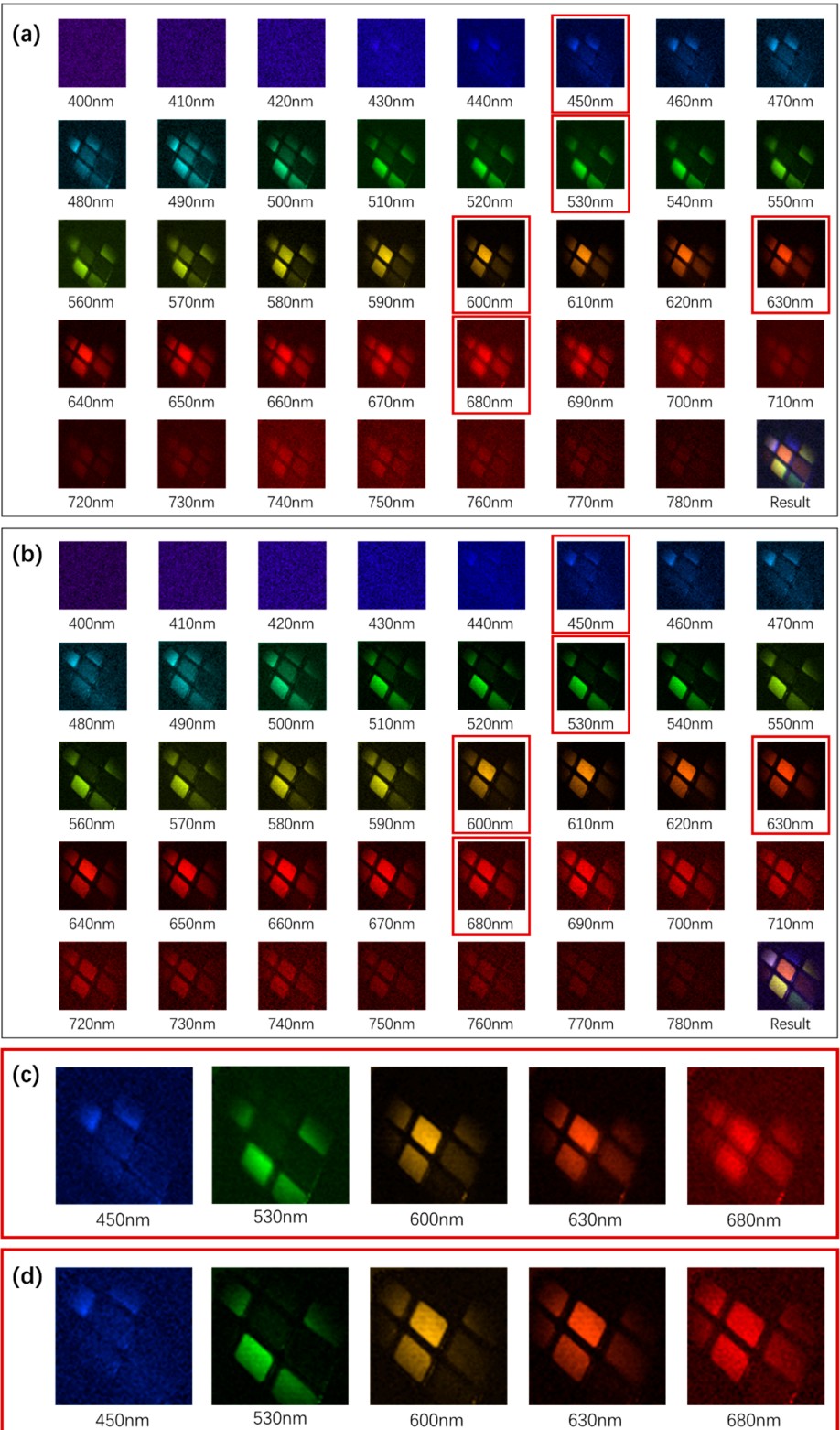

**Figure 7.** Reconstructed hyperspectral images over 400–780 nm in the control (**a**) and TV lens (**b**) groups, with their composite-colored images in the last figure. (**c**,**d**): Enlargements of the respective images in the red boxes of (**a**,**b**) reveal that sharp edges are only seen in the yellow bands of the single-lens system, while they are distinct over the vast majority of band range of the TV system.

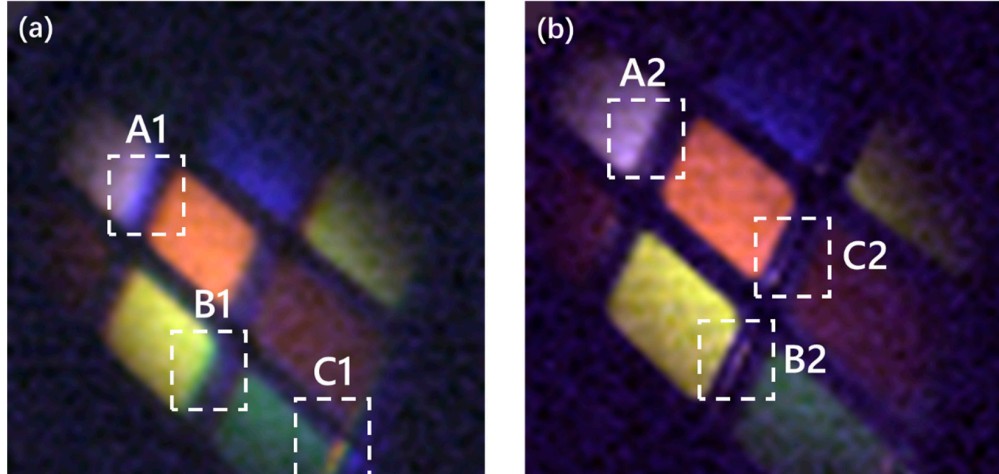

**Figure 8.** Reconstructed colored images of the control group (**a**) and the TV lens group (**b**). The white boxes show the main differences: purple or yellow haloes appear at the edges of the color blocks in (**a**), while in (**b**), even details such as the white edges in boxes B2 and C2 are clearly revealed.

In order to quantify the performance of the two schemes, we examine a cross-section profile in the two resultant images (shown by the red lines in Figure 9a,c). The pixels' gray values on the red line in each spectral band of 400–780 nm are successively extracted from left to right and plotted in the form of line charts in Figure 9b,d. Among them, six spectral bands (440 nm for blue, 490 nm for cyan, 530 nm for green, 580 nm for yellow, 630 nm for orange, and 680 nm for red) are selected. In the two sets of figures, three groups of pixels ①, ②, and ③ at the object's edge positions are chosen where there are marked discontinuous changes in the gray values. Group ① corresponds to the edge of the white block and the maximum gray value point in the region, while group ② in the center of the black gap corresponds to the minimum gray value. Group ③ in the TV lens group of Figure 9c corresponds to the reflected white edge and the minimum gray value in the region, but in the control group of Figure 9a, this detail is too blurred, so the edge of the orange block is chosen, which corresponds to the maximum gray value in this region. In Figure 9b,d, each group is labelled and connected across the spectral bands with a blue line, while the vertical red dotted lines denote the abscissa of the chosen points with the first band (440 nm) as the reference origin. As the wavelength increases in Figure 9b, the corresponding points in the same group move to the left due to lateral chromatism. Additionally, the distance between groups ① and ② is significantly different in the change from blue to red; that is, from 3 pixels apart in the blue band to only 1 pixel apart in the red band, which also indicates the existence of lateral chromatism. However, in Figure 9d, we can see that the blue lines connecting the groups are straighter and almost coincide with the red lines, showing that aberration is corrected and variation with wavelength diminished. The overall deviations of the three positions from the reference origin point, as a function of wavelength, are shown in Figure 10a,b.

In order to quantify the chromatic aberration, we calculated the pixel column number variances of the deviation at the three positions, as shown in Table 1. It can be seen that the variances in the TV lens scheme are on average more than an order of magnitude smaller than those in the single-lens scheme.

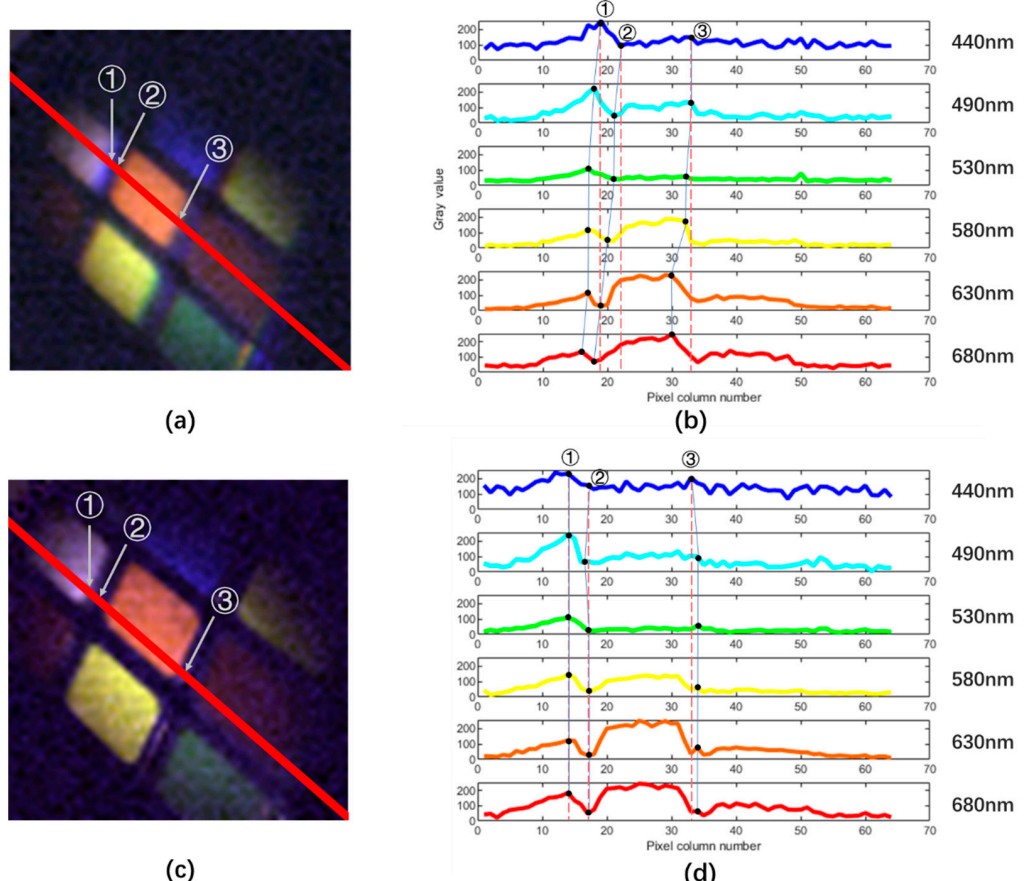

**Figure 9.** Comparison of the performance of chromatic aberration correction. The intensity profiles across the red lines in the reconstructed images of (**a**) the control group and (**c**) the TV lens group vs. pixel number for different wave bands are shown in (**b**,**d**). Three groups of pixels ①, ②, and ③ are selected and in (**b**,**d**) are connected across the spectral bands with a blue line, while the vertical red dotted lines denote the abscissa of the chosen points with the first band (440 nm) as the reference origin.

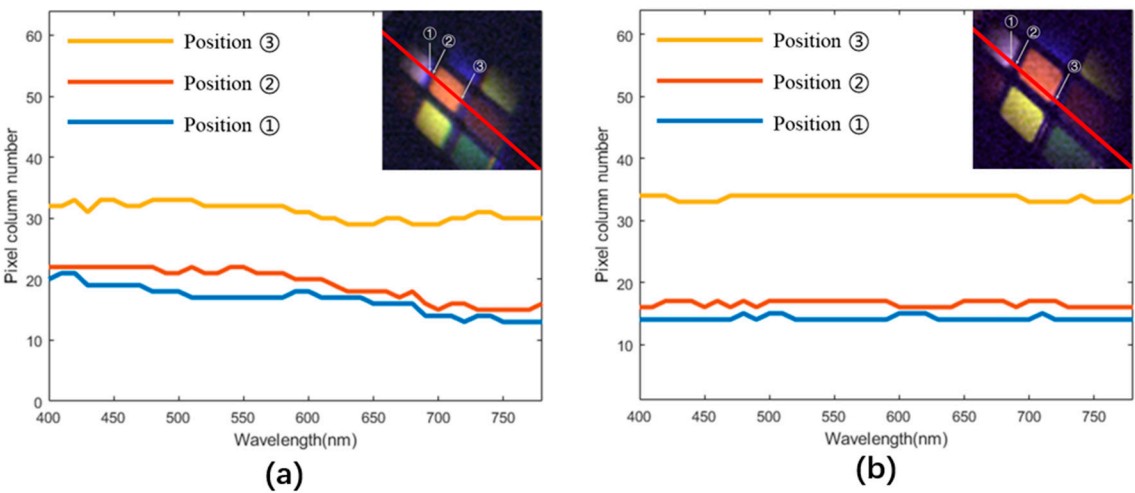

**Figure 10.** Pixel column number of the ①, ②, and ③ positions vs. wavelength over 400–780 nm for the control group (**a**) and the TV lens group (**b**). A flat line indicates no deviation in the focal point position, i.e. no lateral chromatic aberration occurs.

**Table 1.** Position Coordinate Variance in the Two Schemes.

| Scheme Name | ① | ② | ③ |
|---|---|---|---|
| HSPI scheme | 4.8889 | 7.1137 | 1.8659 |
| CHSPI scheme | 0.1473 | 0.2459 | 0.2025 |
| Improvement (HSPI/CHSPI) | 33.1901 | 28.9292 | 9.2143 |

The theoretical spatial resolution limits of conventional imaging and the achieved resolutions of the two SPI systems are compared below to quantify their performance from another perspective. The apertures $D_C$ and $D_T$ of the single lens and TV lens are, respectively, 30 mm and 16 mm. The smallest unit that was clearly resolved by the single lens system was the 4 mm width of the black gaps between the colored blocks in A1 of Figure 8a, whereas the TV lens system could clearly resolve the white reflective strips in B2 and C2 of Figure 8b, which have a width of less than 1 mm. The distance between the object and the lens is more than 45 cm. Thus, the theoretical and actual angular resolutions can be derived as follows.

For a central wavelength $\lambda$ of 550 nm the theoretical resolution limit of conventional imaging using a single-lens system is

$$\varphi_C = 1.22\lambda/D_C = 1.22 \times 550 \text{ nm}/30 \text{ mm} = 0.022 \text{ mrad} \tag{13}$$

while that of the TV lens system is

$$\varphi_T = 1.22\lambda/D_T = 1.22 \times 550 \text{ nm}/16 \text{ mm} = 0.042 \text{ mrad} \tag{14}$$

The spatial resolution actually achieved in the single-lens SPI setup was

$$\varphi_T{}' \approx \Delta l/L = 4 \text{ mm}/450 \text{ mm} = 8.9 \text{ mrad} \tag{15}$$

while that in the improved SPI system was

$$\varphi_T{}' \approx \Delta l/L = 1 \text{ mm}/450 \text{ mm} = 2.2 \text{ mrad} \tag{16}$$

Above are merely the results of our recent proof-of-principle experiments at present, and the actual spatial resolution limits need to be explored in subsequent research. Although the current measured resolutions are far from the theoretical values, comparing Equations (15) and (16), we can see that the achieved spatial resolution of the optimized TV lens system is more than a factor of four better than that of the control setup, even though its aperture is much smaller.

## 7. Conclusions

As the first reported demonstration of chromatic aberration correction in hyperspectral SPI, our scheme succeeded in achieving an order of magnitude improvement in image quality across a wide spectral range of 400–780 nm. The images are reconstructed via an improved adaptive and iterative correction single-pixel algorithm, which allows information across all bandwidths to be retrieved within a short period of time with an image quality enhancement of more than 3 dB on average. Our scheme could provide a technical solution for spectral de-aliasing in SPI settings for microscopy and general analysis while providing a compact, lightweight, stable solution for remote sensing and satellite applications.

**Author Contributions:** Conceptualization, Y.L.; methodology, Y.L.; software, Y.L.; validation, Y.L., M.-Y.S., and Z.-H.Z.; investigation, Y.L.; data curation, Y.L.; writing—original draft preparation, Y.L.; writing—review and editing, Z.-H.Y., Y.-J.Y., and L.-A.W.; visualization, Y.L.; funding acquisition, Z.-H.Y., Y.-J.Y., and L.-A.W. All authors have read and agreed to the published version of the manuscript.

**Funding:** This work was supported in part by the National Natural Science Foundation of China under Grants 61973018, 62173039, and 61975229, the Civil Space Project under Grant D040301, the National Key Research and Development Program of China under Grant 2018YFB0504302, and the Defense Industrial Technology Development Program under grant JCKY2021602B036.

**Institutional Review Board Statement:** Not applicable.

**Informed Consent Statement:** Not applicable.

**Data Availability Statement:** Not applicable.

**Conflicts of Interest:** The authors declare no conflict of interest.

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
