# Peer review of "Chromatic-Aberration-Corrected Hyperspectral Single-Pixel Imaging"

_photonics, doi:10.3390/photonics10010007_

Round 1

Reviewer 1 Report

Dear authors,

you have presented an advance technique on correcting single-pixel images for their chromatic aberration, which improves significantly the analysed hyperspectral image.

I have two major questions to address you with regards your proposed methodology:

1) How fast is your analysis when compared to traditional techniques, like the ones you mentioned in the introduction - on the same processing software/hardware for comparability reasons?

2) Could an optical characterisation of the sensor be advantageous as well? A similar one to the point spread function for typical pixel-based detectors when illuminating with panchromatic light source? What would be the complications/merits of such an approach?

Minor comments

Line21: SPL acronym not given earlier

Line 67: I think is worth mentioning here why you got a double gaussian structure and point the reader to line 146 to justify your choice.

lines 100, 101: maybe word has not inserted any symbols in M(x) ad I(x), as I can only see empty boxes?

lines 180, 181: I suggest using 3x3x3 and 54x54x54 mm3

Thank you.

Author Response

Response to Reviewer 1 Comments

Dear reviewer:

Thank you for your comments concerning our manuscript entitled “Chromatic Aberration Corrected Hyperspectral Single-Pixel Imaging” (photonics-2028491). Your comments are all very helpful for improving our paper, and accordingly, we have made several revisions. Our detailed responses to your comments (in blue), the actions taken (in bold black) and the text change in the paper (in red) are listed below. All the revisions to the manuscript have been marked up using the “Track Changes” function.

Once again, we thank you for your valuable comments and suggestions, and hope that our revised manuscript is now suitable for publication.

                                                                                        Yours sincerely,             

                                                                                             Liu Ying    

Reviewer 2 Report

The achromatic design in single pixel spectral camera has been proposed in this manuscript. This is an important issue to be considered in the design of the optical imaging systems. An optimal reconstruction method was also described. The experiment result verified the feasibility of the optimal design. Overall, I think the proposed work is meaningful and suitable for publishing on this journal. There are some suggestions for the authors to improve the quality of the manuscript.

(1) Some sentences in the paper are too long to understand readily;

(2) In the performance evaluation, an image evaluation metric, CSNR, has been used. Some optical spectrum related metrics such SAM should be introduced to evaluate the performance. 

Author Response

(The authors gave the same response as above.)

Author Response

(The authors gave the same response as above.)

Round 2

Reviewer 3 Report

Dear Authors,

I have greatly appreciated your response to my comments, and I think that your modifications have indeed improved the overall quality of the article. However, I still stand by my opinion of the original review.

The major problem for me is the optimization section (which I do not consider an optimization), and how much importance is given to it. 

Author Response

The Response Letter to Reviewer 3

Dear Reviewer 3:

Thank you for your letter and comments concerning our manuscript entitled “Chromatic Aberration Corrected Hyperspectral Single-Pixel Imaging” (photonics-2028491). Your comments are all very helpful for improving our paper, and accordingly, we have made several revisions. Our detailed responses to your comments (in blue), the actions taken (in bold black) and the text change in the paper (in red) are listed in the MS word file. All the revisions to the manuscript have been marked up using the “Track Changes” function. The main revisions and our response to your comments are as follows:  

Comment 1: The major problem for me is the optimization section (which I do not consider an optimization), and how much importance is given to it.

Reply: Thank you for your valuable comments. We have carefully reflected on the innovation of the algorithm part and found that in practice, it does apply more to “correction” than to our previous “optimization”. In this way, we have corrected the wording from "optimization" to "correction" at lines 69, 203, 208, 213, 293 and in the caption of Fig. 6.

As a matter of fact, this algorithm improvement is quite necessary for the single-pixel imaging field, which have been mentioned in the last letter “We have discussed with peers for many times and found that similar errors always appear in the upper left corner of the image recovered during the actual sampling of single-pixel imaging. Through mathematical derivation, we can conclude that this area will gather the global noise of the image. However, it never appears in computational simulations. The reason is that the randomness of noise cannot be completely predicted during simulation (the variance of actual noise is not 0, while white noise with a variance of 0 is usually adopted during simulation). This is a common problem in computational single-pixel imaging”. Therefore, the improvement of this part would play a much greater role in the practical application than it might seem to do.

Once again, we thank you very much for your comments and suggestions and hope that our revised manuscript is now suitable for publication.

Sincerely,

Zhao-Hua YANG